# Associations between School Lunch and Obesity in Korean Children and Adolescents Based on the Korea National Health and Nutrition Examination Survey 2017–2019 Data: A Cross-Sectional Study

**DOI:** 10.3390/nu15030698

**Published:** 2023-01-30

**Authors:** Yeji Kim, Kumhee Son, Jieun Kim, Miji Lee, Kyung-Hee Park, Hyunjung Lim

**Affiliations:** 1Department of Medical Nutrition, Graduate School of East-West Medical Science, Kyung Hee University, Yongin 17104, Republic of Korea; 2Research Institute of Medical Nutrition, Kyung Hee University, Seoul 02447, Republic of Korea; 3Department of Family Medicine, Hanllym University sacred Heart Hospital, Anyang 14068, Republic of Korea

**Keywords:** school lunch, children, adolescents, obesity, Korea

## Abstract

Obesity in children and adolescents is a serious global problem. In Korea, approximately 35% of students’ daily nutrient intake is from school lunch (SL), and all schools provide SL. However, the association between SL and obesity remains controversial. This study examined this association and the daily nutrient intake according to lunch type in Korean children and adolescents. We analyzed 1736 individuals aged 7–18 from the Korea National Health and Nutrition Examination Survey (2017–2019), a cross-sectional study, using logistic regression analysis with odds ratios and 95% confidence intervals. The SL group had higher energy and greater phosphorus, potassium, vitamin A, carotene, vitamin B1, and niacin intake than the non-school lunch (NSL) and skipping lunch (SKL) groups. Protein intake was also higher in the SL group than in the NSL group. The SKL group had higher saturated fatty acid intake, and was thereby 2.5, 1.9, and 2.5 times more likely to have obesity, overweight and obesity, and central obesity (*p* = 0.0071, 0.0459, 0.0092), respectively, than the SL group. Therefore, the SL group consumed more appropriate nutrients than the NSL and SKL groups, and was less likely to become obese than the SKL group. More in-depth prospective studies are needed to elucidate the causal relationship between SL and obesity.

## 1. Introduction

The prevalence of obesity in childhood and adolescence has been increasing worldwide for several decades, with a sharp increase of over 11% in the last 6 years [1]. Likewise, the prevalence of both overweight and obesity among Korean children and adolescence in 2021 was 30.8%, which also implies a sharp increase of approximately 8% in the last 5 years. In particular, the increased rate of obesity was found to be about three times higher than the increased rate of overweight, indicating that obesity in children and adolescents is a very serious social problem in South Korea [2]. Obesity in children and adolescents is influenced by various lifestyles, including dietary habits such as unbalanced diet, irregular diet, overeating, and skipping meals [3,4]. According to preliminary surveys of Korean children and adolescents, negative dietary habits such as fast food/beverage intake and skipping breakfast increased, while positive dietary habits such as vegetable and dairy intake and physical activity decreased [2]. Childhood and adolescence are the stages that dietary habits are established; if the established dietary habits at this time are incorrect, obesity might occur [5].

School is where children and adolescents spend most of their daytime and where their dietary habits and lifestyle are formed. School lunch (SL) not only has a role in serving food but also functions as an important educational method for school children to acquire proper dietary habits and knowledge [6]. However, policies on school meals differ from country to country [7]; hence, the impact of school meals on children and adolescents is inevitably different [8]. SL programs have helped reduce obesity rates [9,10,11], improve mental health [12], food insecurity, and poor health [13], and are also, surprisingly, associated with higher future income [14]. However, some studies showed that eating school meals negatively affects obesity [15,16] and contributes to high intake of saturated fat and sodium [17].

In Korea, 11,903 schools, including elementary, middle, high schools, and special schools, allow SL, and 99.9% of their students take SL [18]. Although all schools in Korea are implementing the SL system, studies on the effect of SL on health are currently limited. Therefore, this study aimed to examine the daily nutrient intake according to lunch type and the association between SL and obesity among children and adolescents in South Korea, using data from the 2017–2019 Korean National Health and Nutrition Survey (KNHANES).

## 2. Materials and Methods

### 2.1. Data Source and Study Population

Data was obtained from KNHANES VII (2017–2018) and VIII (2019). KNHANES is a nationwide, population-based, cross-sectional survey conducted by the Korea Centers for Disease Control and Prevention since 1998 in South Korea. This survey is based on data from Statistics Korea’s Population and Housing Census and the Public Price of Apartment Houses. A total of 282 households in 192 regions are selected as probability samples for Koreans over 1 year of age, and representative groups are selected as sample designs stratified by region, age, and sex. Two-stage stratified cluster sampling and rolling sampling surveys were performed to represent all parts of the nation. Therefore, KNHANES is valuable in terms of representativeness and reliability. KNHANES consists of a health examination, interview, and nutrition survey. Questionnaires are selected by collecting opinions from the central government and experts in related fields, considering the items necessary for health policy establishment in Korea. Skilled medical staff and interviewers conducted the survey. They are trained for 1 month and then sent to the survey site. Afterward, their ability to conduct investigations is continuously verified through regular training (7 times a year) and on-site quality control. Health examinations are composed of anthropometric measurements, blood chemistry tests, and urinalysis. A health interview is self-reported; it assesses physical activity, sleep health, weight control, drinking status, and quality of life. Both the health examinations and interview are conducted at a mobile examination center with rooms for surveys and health examination instruments. Moreover, a nutrition survey, which comprises dietary habits and food intake, is conducted at participants’ homes a week after the health interview [19].

Of the 24,229 participants, those aged 7–18 years were enrolled in this study. The exclusion criteria were as follows: not within 7–18 years (*n* = 21,152), extremely low or high caloric intake (<500 kcal/day or >5000 kcal/day) (*n* = 387), no anthropometric data (*n* = 191), and who took the nutrition survey on Sunday or Monday (*n* = 763). Dietary intake was assessed using the 24 h dietary recall method; thus, surveys conducted on Sunday or Monday indicated that assessments were performed on weekends when SL was unavailable. Finally, 1736 participants were included in our study (Figure 1). Informed consent was obtained from all participants at the time of the survey, and Kyung Hee Institutional Review Board approved this study protocol (KHGIRB-22-072).

### 2.2. Assessment of Sociodemographic Data

Participants’ sociodemographic data included age, sex, school grade, household income, residential area, and physical activity. Those aged 7–18 years were included, and school grade was categorized into elementary (7–12 years), middle (13–15 years), and high school (16–18 years). Household income was categorized into low, low-middle, middle-high, and high. The residential area was classified into rural (dong) and urban (eup/myeon). Physical activity was defined as ≥60 min of physical activity and muscular exercise. However, physical activity was assessed only in children aged ≥ 12 years in the KNHANES. Therefore, physical activity data were only analyzed in participants aged ≥ 12 years.

### 2.3. Anthropometric Measurement

Using standardized techniques and calibrated equipment, the professional medical staff measured the participants’ weight (GL-6000-20; Caskorea, Seoul, Republic of Korea), height (Seca 225; GmbH & Co. KG, Hamburg, Germany), and waist circumference (WC) (Seca 201; GmbH & Co. KG, Hamburg, Germany). Body mass index (BMI) was calculated as weight (kg) divided by height (m^2^).The age- and sex-specific BMI percentiles of participants were analyzed according to the 2017 Korean National Growth Charts [20]. The weight status based on the BMI percentiles was as follows: (a) underweight, BMI < 5th percentile; (b) normal weight, 5th percentile ≤ BMI < 85th percentile; (c) overweight, 85th percentile ≤ BMI < 95th percentile; and (d) obesity, BMI ≥ 95th percentile. Furthermore, the participants’ WC percentile was analyzed according to the National Cholesterol Education Program Adult Treatment Panel III. Central obesity was diagnosed according to the 2007 Korean Children and Adolescents Growth Standard [21]. The WC status based on the WC percentile was as follows: (a) central obesity, WC ≥ 90th percentile; and (b) non-central obesity, WC < 90th percentile.

### 2.4. Dietary Intake and Lunch Type

Trained interviewers assessed participants’ dietary intake using the 24 h dietary recall method. In the nutrition survey, a trained nutritionist visited individual households to investigate their eating habits, food type consumed, and food intake 1 day before the survey. To increase the accuracy of the intake survey, interviewers used aids such as two-dimensional food containers, food models, measuring cups, measuring spoons, thicknesses, 30 cm ruler, and tape measure. For energy and 23 nutrients such as carbohydrates, fat, protein, dietary fiber, calcium, phosphorus, iron, sodium, potassium, vitamin A, carotene, retinol, vitamin B1, vitamin B2, vitamin C, niacin, cholesterol, saturated fat, monounsaturated fatty acids, polyunsaturated fatty acids, *n*-3 fatty acids, *n*-6 fatty acids, and sugars, the daily intake was calculated using data from the Korean Foods and Nutrients Databases of the Rural Development Administration. Moreover, participants were categorized into three groups according to lunch type: (a) SL group, having lunch at school; (b) non-school lunch (NSL) group, having lunch outside of school; and (c) skipping lunch (SKL) group, not having lunch anywhere.

### 2.5. Statistical Analysis

All statistical analyses were performed using PROC SURVEY with SAS version 9.4 (SAS Institute, Cary, NC, USA). Continuous variables such as age, height, and weight were presented as mean ± standard error (SE) according to the PROC SURVEYMEANS procedure and were adjusted for age and sex, tested with the Bonferroni post–hoc multiple comparison test after one-way analysis of variance and analysis of covariance using the SURVEYREG procedures. Categorical variables such as sex, school level, residential area, household income level, and physical activity were expressed as frequencies (n) and percentages (%) according to the PROC SURVEYFREQ procedure. Nutrient intake among the SL, NSL, and SKL groups was compared using a multivariate analysis of variance adjusted for age and sex. We also conducted a logistic regression analysis with odds ratios (ORs) and 95% confidence intervals (CIs) to assess the association between SL and obesity according to the PROC SURVEYLOGISTIC procedure. The prevalence of obesity, overweight and obesity, and central obesity according to lunch type was examined using a non–adjusted multivariate logistic regression model (Model 1). A second model, Model 2, was used after adjusting for sex and age, and a third, Model 3, after adjusting for sex, age, household income, and residential area. A *p*-value < 0.05 was considered statistically significant.

## 3. Results

Table 1 shows the general characteristics of the participants according to lunch type. The mean ages of the SL (*n* = 1168, 63.2%), NSL (*n* = 493, 31.5%), and SKL (*n* = 75, 5.3%) groups were 11.8 ± 0.1, 13.8 ± 0.2, and 14.5 ± 0.5 years, respectively (*p* < 0.0001). The distribution of school levels was significantly different between the three groups (*p* < 0.0001). In the SL group, the majority were in elementary (59.5%), followed by middle school (22.4%) and high school (18.1%). Most students in the NSL group were in high school (41.7%), followed by elementary (36.1%) and middle school (22.2%), consistent with the SKL group (50.3%, 26.2%, and 23.5%, respectively). No significant differences were noted between these groups regarding the residential area and household income.

Table 2 shows the anthropometric measurements and obesity classifications according to lunch type. The distribution of obesity classification significantly differed between the three groups (*p* = 0.021). While 69.2%, 69.1%, and 56.9% of the participants in the SL, NSL, and SKL groups had normal weight, obesity accounted for 13.7%, 13.5%, and 30.4%, respectively. Furthermore, overweight was found in 8.7%, 8.2%, and 5.8%, and underweight was noted in 8.5%, 9.3%, and 6.9%, respectively. The proportion of participants with central obesity also significantly differed between these groups (SL, NSL, and SKL: 11.3%, 12.7%, and 27.2%, *p* = 0.004). Thus, the SKL group had the highest proportion of participants with obesity (30.4%) and central obesity (27.2%).

Table 3 shows the daily nutrients intake of the participants according to lunch type. The SL group had a significantly higher intake of energy (2015.0 ± 28.1 vs. 1726.7 ± 113.5 kcal/day), carotene (1986.0 ± 59.4 vs. 1383.9 ± 172.5 μg/day), niacin (12.5 ± 0.2 vs. 9.7 ± 0.8 mg/day) and vitamin B1 (1.4 ± 0.0 vs. 1.0 ± 0.1 mg/day) than that of the SKL group. The SL group also had a significantly higher intake of energy (2015.0 ± 28.1 vs. 1943.6 ± 44.2 kcal/day), protein (74.4 ± 1.2 vs. 69.2 ± 1.9 g/day), phosphorus (1088.0 ± 16.5 vs. 995.1 ± 24.1 mg/day), potassium (2410.1 ± 38.7 vs. 2165.9 ± 51.4 mg/day), carotene (1986.0 ± 59.4 vs. 1653.2 ± 83.6 μg/day), vitamin B1 (1.4 ± 0.0 vs. 1.3 ± 0.0 mg/day) and vitamin A (407.2 ± 12.7 vs. 326.1 ± 14.3 μgRE/day) than the NSL group. Conversely, the SKL group had a significantly higher intake of saturated fatty acid (SFA) (19.5 ± 0.4 vs. 19.8 ± 2.2 g/day) than the SL group.

Table 4 shows the association of lunch type with obesity, overweight and obesity, and central obesity. Model 3 was adjusted for sex, age, household income, and region, which were the confounding factors for obesity in childhood and adolescence. Compared with the SL group, the SKL group had 2.5 times higher risk for obesity (OR = 2.479; 95% CI = 1.282–4.791, *p* = 0.0071), 1.9 times higher risk for overweight and obesity (OR = 1.861; 95% CI = 1.011–3.426, *p* = 0.0459), and 2.5 times higher risk for central obesity (OR = 2.456; 95% CI = 1.251–4.824, *p* = 0.0092).

## 4. Discussion

This study investigated the daily nutrient intake according to lunch type and the associations between lunch type and obesity in South Korean children and adolescents, using data representing the whole country. Compared with the NSL and SKL groups, the SL group consumed nutrients such as energy, protein, and most of the micronutrients more adequately. The SKL group had a lower energy intake but a higher SFA intake. Thus, the SKL group was more likely to have obesity or central obesity than the SL group.

SL reportedly increases the risk of obesity [15,16,22]. Some studies in the United States found that the SL menus mainly consist of fast foods such as chicken, hamburgers, tacos, burritos, milk, and oil or fat. Thus, given that the SL group is more likely to consume excess fat, saturated fat, and other nutrients, they were at high risk of obesity [23,24,25]. Another study conducted in other Western countries also has fast foods as the main menu for SL [26]. Therefore, the intake of total fat and saturated fats is also high. Consequently, having a high-fat and high-SFA lunch may negatively affect daily intake, increasing the risk of obesity. Moreover, children receiving free or reduced-price SLs through the National School Lunch Program (NSLP) tend on average to have worse health outcomes, such as food insecurity, poor health, and obesity, than those who do not participate in this program, because of higher calorie intake at lunchtime [16]. However, selection bias is possible because NSLP participation is inherently targeted at children with a high risk for obesity from low-income households [13].

Our study results differ from those of studies in Western countries. In the present study, although the SL group had higher intakes of energy and several nutrients than those of NSL and SKL groups, when compared with the Dietary Reference Intakes for Koreans (KDRIs), the SL group’s intake of energy, protein, and most micronutrients was found to be the most appropriate compared to those of the NSL and SKL groups [27]. SL plays an important educational role not only in eating lunch but also in acquiring proper eating habits and knowledge [6]. According to a previous study, the SL group consumed most nutrients more adequately than the other groups because SL was provided according to strict guidelines [28]. Two studies showed evidence that the provision of SL had a positive effect on the formation of appropriate eating habits and that receiving free and reduced-price meals improved children’s health outcomes [10,29]. SL provision was effective in establishing healthy eating habits, including regular meals, avoiding unhealthy meals, choosing appropriate snacks, and receiving adequate nutrition [29]. Therefore, SL can help prevent obesity by not only supplying sufficient nutrients but also by changing the knowledge and perception of healthy eating habits to change overall eating habits. Ultimately, SL may have a role in improving health and preventing obesity in children and adolescents.

However, the risk of obesity did not differ between the NSL group and the SL group. Similar to our study, the previous study also found no association between diet type (SL vs. NSL) and obesity or weight status because the quality of school meals and the composition of the students who consume them have significantly changed over time [30]. However, another study found that high-quality school meal intake was associated with a long-term reduction in obesity rates [9]. Although it is challenging to determine the exact causal association between obesity and meal type because diverse environmental factors are involved in the development of obesity, further study must be conducted to clearly understand the role of SL compared with NSL in reducing the risk of obesity.

In the present study, the SKL group had higher obesity prevalence than the SL group, despite having daily energy intake. The reasons were as follows. First, considering that ghrelin secreted during fasting induces eating behavior, SKL is closely associated with weight gain [31]. Hence, overeating after SKL may cause weight gain. Further, children who skipped lunch often consume low-quality foods, such as ramen, soft drinks, chocolate, chicken, and pizza [32]. These foods have a high content of calories, simple sugars, and fats, and they also have a negative effect on the appetite-related nervous system [33]. Therefore, SKL-induced overeating during dinner can cause obesity [34]. Second, irregular eating habits such as meal skipping affect weight gain. SKL can lead to the insufficient intake of essential nutrients because it reduces the intake of vegetables, fruits, whole grains, dairy products, vegetable proteins, and seafood [35]. Insufficient nutrient intake can cause hunger even after taking enough calories [36]. If an irregular eating habit continues, the basal metabolic rate decreases. In preparation for irregular energy intake, nutrients are converted and stored into adipose tissues, causing obesity [37]. Third, fat intake, particularly excessive SFA, is associated with obesity. The SFA ratio of the SL and SKL groups was 8.7% and 10.3%, respectively, which exceeded the appropriate SFA ratio based on Dietary Reference Intakes for Koreans (KDRIs), that is, 8% [27]. However, the ratio was highest in the SKL group; thus, SKL may be related to obesity. In a previous study, the standard SFA ratio was 10%, but the SL group’s SFA ratio was 12.7% [28]. In another previous study using data from KNHANES between 2016 and 2020, the rate of inappropriate SFA consumption increased from 44.5% to 51.9% in boys and from 47.9% to 55.7% in girls. Similar trends were observed in individuals aged 10–18 years in other countries [38]. This finding is attributed to the fact that more children and adolescents consume fast foods containing high levels of fat, SFA, and sodium annually [39]. In addition, people with obesity or overweight have higher disinhibition scores than those with normal weight. Disinhibition reflects overeating and opportunistic tendencies in an obesogenic environment, and a high score indicates a higher preference for foods, particularly sweet and high-fat foods [40]. Therefore, dietary restrictions, such as skipping meals, may lead to weight gain over time by increasing disinhibition and overeating in the presence of sufficient foods [41].

A school is a place where the lifestyle habits of children and adolescents are established. Given that approximately one-third of the daily intake is generally consumed in this institution, it plays a central role in preventing and managing childhood obesity. Therefore, the policy on SLs should be promoted at the governmental level [5]. In particular, in order to promote intersectoral policies for the enactment of relevant laws for the implementation and quality maintenance of school meals, the training of manpower such as nutritionists in charge of catering and development of educational programs, and securing budget for policy implementation and facility expansion, the central government must work together with the Ministry of Education, Health, and Welfare [42].

This study has several strengths. First, it is the first to investigate the relationship between SL and obesity in children and adolescents in South Korea, where the school meal service rate is high. Second, it used KNHANES data, which are representative sample data of a national survey targeting South Koreans. We minimized the risk of sampling error by using a complex multilevel stratified cluster sampling method. However, this study also had some limitations. First, dietary intake was assessed using a single 24 h dietary recall, which may not represent the overall intake of the children and adolescents. However, at the population level, this method can provide rich details about mean dietary intake for a given day. Second, the study is cross-sectional; hence, elucidating the causative relationship between SL and obesity is difficult.

## 5. Conclusions

Students availing SL had a more appropriate nutrient intake than the NSL and SKL groups. Compared with the SL group, the SKL group was at 2.5, 1.9, and 2.5 times higher risk of developing obesity, overweight and obesity, and central obesity, respectively. The causal relationship between SL and obesity cannot be proven because other lifestyle influences cannot be ruled out. However, considering that children taking SL consumed more nutritious meals than those who did not and SKL showed an association with childhood obesity, lunch intake should be encouraged.

Additional longitudinal studies focusing on the cause–effect relationship between SL intake and obesity are needed to obtain results that complement the study limitations.

## Figures and Tables

**Figure 1 nutrients-15-00698-f001:**
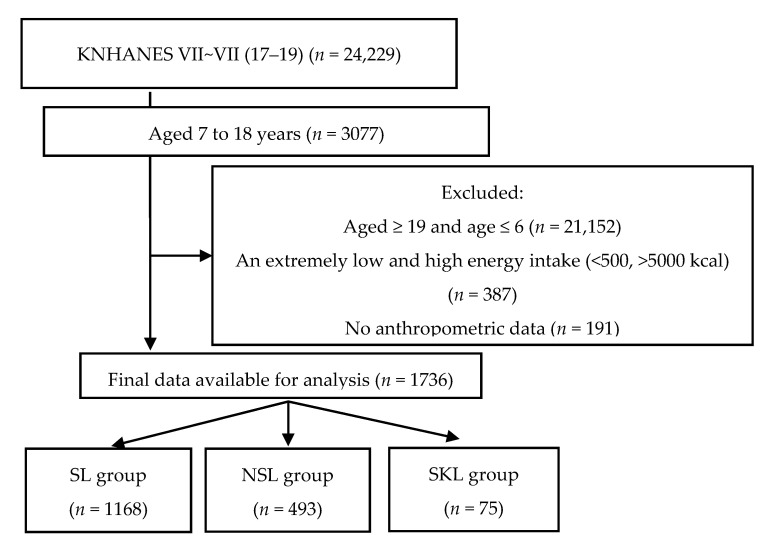
Flowchart of this study. KNHANES, Korean National Health and Nutrition Survey; SL, school lunch; NSL, non-school lunch; SKL, skipping lunch.

**Table 1 nutrients-15-00698-t001:** Sociodemographic characteristics among children and adolescents in South Korea.

Variables	Total (*n* = 1736, N = 12,161,556)	SL Group (*n* = 1168, N = 7,685,180)	NSL Group (*n* = 493, N = 3,834,876)	SKL Group (*n* = 75, N = 641,501)	*p*-Value ^3^
Age ^1^					
	12.6 ± 0.1	11.8 ± 0.1 ^c^	13.8 ± 0.2 ^ab^	14.5 ± 0.5 ^a^	<0.0001
Sex ^2^					
Boys	896 (51.6)	599 (50.8)	259 (52.5)	38 (55.2)	0.747
Girls	840 (48.4)	569 (49.2)	234 (47.5)	37 (44.8)	
School level ^2^					
Elementary	1067 (50.3)	798 (59.5)	244 (36.1)	25 (26.2)	<0.0001
Middle	361 (22.4)	237 (22.4)	102 (22.2)	22 (23.5)	
High	308 (27.2)	133 (18.1)	147 (41.7)	28 (50.3)	
Residential area ^2^					
Urban	1482 (87.6)	993 (87.2)	424 (88.4)	65 (88.4)	0.882
Rural	254 (12.4)	175 (12.8)	69 (11.6)	10 (11.6)	
Household income level ^2^					
Low	155 (9.5)	116 (10.7)	33 (7.2)	6 (8.8)	0.422
Low-middle	461 (26.3)	308 (26.5)	137 (26.8)	16 (19.9)	
Middle-high	590 (33.2)	402 (32.7)	158 (32.7)	30 (41.8)	
High	529 (31.1)	342 (30.1)	164 (33.3)	23 (29.5)	
Physical activity					
≥60 min of physical activity ^2^					
<5 day/week	750 (93.3)	440 (94.8)	260 (91.6)	50 (90.8)	0.262
≥5 day/week	58 (6.7)	30 (5.2)	24 (8.4)	4 (9.2)	
Muscular exercise ^2^					
<3 day/week	678 (82.4)	402 (84.2)	231 (79.6)	45 (83.3)	0.351
≥3 day/week	130 (17.6)	68 (15.8)	53 (20.4)	9 (16.7)	

SL, school lunch; NSL, non-school lunch; SKL, skipping lunch*; n*, *u*nweighted sample size; N, weighted sample size. ^1^ Continuous variables are expressed as least square mean ± standard error. ^2^ Categorical variables are presented as number (weighted %). ^3^
*p*-values were calculated using the PROC SURVEYREG adjusted for sex with analysis of covariance. Values with different superscript letters represent the results of the post-hoc test (Bonferroni).

**Table 2 nutrients-15-00698-t002:** Anthropometric measurements according to lunch type among children and adolescents in South Korea.

Variables	Total (*n* = 1736, N = 12,161,556)	SL Group (*n* = 1168, N = 7,685,180)	NSL Group (*n* = 493, N = 3,834,876)	SKL Group (*n* = 75, N = 641,501)	*p*-Value
Height (cm) ^1^	153.8 ± 0.5	151.1 ± 0.6 ^b^	157.7 ± 0.9 ^a^	162.4 ± 2.1 ^ab^	0.012
Weight (kg)	49.3 ± 0.6	46.8 ± 0.6	52.8 ± 1.0	59.7 ± 2.7	0.041
BMI (kg/m^2^) ^1^	20.2 ± 0.1	19.8 ± 0.2	20.6 ± 0.3	22.1 ± 0.7	0.113
Underweight ^2,3^	154 (8.6)	98 (8.5)	50 (9.3)	6 (6.9)	0.021
Normal weight	1184 (68.5)	800 (69.2)	340 (69.1)	44 (56.9)	
Overweight	156 (8.4)	109 (8.7)	41 (8.2)	6 (5.8)	
Obesity	242 (14.5)	161 (13.7)	62 (13.5)	19 (30.4)	
WC (cm) ^1^	68.0 ± 0.4	66.6 ± 0.4	69.8 ± 0.7	73.7 ± 1.9	0.264
Normal ^2,4^	1526 (87.4)	1031 (88.7)	437 (87.3)	58 (72.8)	0.004
Central obesity	210 (12.6)	137 (11.3)	56 (12.7)	17 (27.2)	

SL, school lunch; NSL, non-school lunch; SKL, skipping lunch; BMI, body mass index; WC, waist circumference; *n*, unweighted sample size; N, weighted sample size. ^1^ Continuous variables are expressed as are mean ± standard error, *p*-values were adjusted for sex and age with analysis of covariance. ^2^ Categorical variables are expressed as numbers (weighted %). ^3^ Obesity categories were divided into BMI percentile: underweight, BMI < 5th percentile; normal weight, BMI 5th–84th percentile; overweight, BMI 85th–94th percentile; obesity, BMI ≥ 95th percentile. ^4^ Central obesity categories were divided into waist circumference (WC) percentile: normal, WC < 90th percentile; central obesity, WC ≥ 90th percentile. *p*-values were calculated using the PROC SURVEYREG adjusted for sex with analysis of covariance. Values with different superscript letters represent the results of the post-hoc test (Bonferroni).

**Table 3 nutrients-15-00698-t003:** Daily nutrients intake according to lunch type among children and adolescents in South Korea.

Variables	SL Group (*n* = 1168, N = 7,685,180)	NSL Group (*n* = 493, N = 3,834,876)	SKL Group (*n* = 75, N = 641,501)	*p*-Value ^2^
Energy (kcal/day)	2015.0 ± 28.1 ^a,1^	1943.6 ± 44.2 ^b,c^	1726.7 ± 113.5 ^c^	0.0001
Carbohydrate (g/day)	301.1 ± 4.3	287.8 ± 6.3	250.1 ± 16.4	0.384
Fat (g/day)	54.9 ± 1.1	54.5 ± 1.8	52.7 ± 5.6	0.022
Protein (g/day)	74.4 ± 1.2 ^a^	69.2 ± 1.9 ^b^	60.1 ± 4.2 ^a,b^	0.0003
C:F:p (%)	60.3:24.0:14.7	59.8:24.8:14.3	60.0:24.9:14.5	
Fiber (g/day)	19.1 ± 0.4	19.2 ± 0.5	16.6 ± 1.5	0.442
Calcium (mg/day)	560.3 ± 10.7	488.2 ± 17.5	476.6 ± 44.9	0.095
Phosphorus (mg/day)	1088.0 ± 16.5 ^a^	995.1 ± 24.1 ^b^	890.6 ± 64.0 ^a,b^	0.002
Iron (mg/day)	10.9 ± 0.2	10.3 ± 0.3	8.7 ± 0.6	0.037
Sodium (mg/day)	2940.8 ± 58.4	2887.4 ± 86.6	2448.7 ± 191.2	0.065
Potassium (mg/day)	2410.1 ± 38.7 ^a^	2165.9 ± 51.4 ^b^	1945.4 ± 133.7 ^a,b^	0.0002
Vitamin A(μgRE/day)	407.2 ± 12.7 ^a^	326.1 ± 14.3 ^b^	286.8 ± 27.9 ^a,b^	0.002
Carotene (μg/day)	1986.0 ± 59.4 ^a^	1653.2 ± 83.6 ^b,c^	1383.9 ± 172.5^c^	0.002
Retinol (μg/day)	241.2 ± 11.5	187.8 ± 13.7	170.4 ± 20.9	0.102
Vitamin B1 (mg/day)	1.4 ± 0.0 ^a^	1.3 ± 0.0 ^b,c^	1.0 ± 0.1 ^c^	<0.0001
Vitamin B2 (mg/day)	1.7 ± 0.0	1.6 ± 0.1	1.5 ± 0.1	0.951
Vitamin C (mg/day)	61.4 ± 2.4	61.0 ± 4.0	52.6 ± 8.0	0.384
Niacin (mg/day)	12.5 ± 0.2 ^a^	11.8 ± 0.3 ^a,b^	9.7 ± 0.8 ^b^	0.029
Cholesterol (mg/day)	276.3 ± 6.5	277.2 ± 17.3	243.3 ± 30.1	0.822
SFA (g/day)	19.5 ± 0.4 ^b^	19.3 ± 0.7 ^a,b^	19.8 ± 2.2 ^a^	0.007
MUFA (g/day)	17.6 ± 0.4	18.0 ± 0.7	17.6 ± 2.1	0.017
PUFA (g/day)	12.6 ± 0.3	12.0 ± 0.4	10.6 ± 1.2	0.946
w-3 FA (mg/day)	1.6 ± 0.1	1.4 ± 0.1	1.2 ± 0.2	0.558
w-6 FA (mg/day)	11.0 ± 0.3	10.5 ± 0.4	9.3 ± 1.0	0.951
Sugar (g/day)	65.7 ± 1.4	66.8 ± 2.4	61.6 ± 5.1	0.016

SL, school lunch; NSL, non-school lunch; SKL, skipping lunch; *n*, unweighted sample size; N, weighted sample size; C:F:P, carbohydrate:fat:protein ratio; SFA, saturated fatty acid; MUFA, monounsaturated fatty acid; PUFA, polyunsaturated fatty acid; w-3 FA, omega-3 fatty acid; w-6 FA, omega-6 fatty acid. ^1^ Continuous variables are expressed as are mean ± standard error. ^2^ *p*-values were calculated using the PROC SURVEYREG adjusted for sex, age and total energy intake with analysis of covariance. Values with different superscript letters represent the results of the post-hoc test (Bonferroni).

**Table 4 nutrients-15-00698-t004:** Odds ratios of obesity according to the lunch type among children and adolescents in South Korea.

Variables		Model 1 ^1^			Model 2 ^2^			Model 3 ^3^		
		OR	95% CI	*p*-Value ^4^	OR	95% CI	*p*-Value	OR	95% CI	*p*-Value
Obesity	SL group	1.000 (ref)			1.000 (ref)			1.000 (ref)		
	NSL group	0.982	(0.696–1.385)	0.916	0.916	(0.642–1.307)	0.627	0.926	(0.650–1.319)	0.670
	SKL group	2.759	(1.450–5.249)	0.002	2.511	(1.307–4.822)	0.006	2.479	(1.282–4.791)	0.007
Overweight and obesity	SL group	1.000 (ref)			1.000 (ref)			1.000 (ref)		
	NSL group	0.958	(0.707–1.298)	0.781	0.921	(0.680–1.247)	0.593	0.944	(0.699–1.273)	0.704
	SKL group	1.967	(1.087–3.559)	0.025	1.861	(1.027–3.372)	0.041	1.861	(1.011–3.426)	0.046
Central obesity	SL group	1.000 (ref)			1.000 (ref)			1.000 (ref)		
	NSL group	1.148	(0.762–1.730)	0.508	1.019	(0.678–1.532)	0.929	1.028	(0.685–1.542)	0.893
	SKL group	2.944	(1.497–5.788)	0.002	2.513	(1.283–4.920)	0.007	2.456	(1.251–4.824)	0.009

SL, school lunch; NSL, non-school lunch; SKL, skipping lunch; OR, odds ratio; CI, confidence interval. ^1^ Model 1 was not adjusted. ^2^ Model 2 was adjusted for sex and age. ^3^ Model 3 was adjusted for sex, age, household income and region. ^4^ *p*-values were the result of multivariate logistic regression analysis or the association of lunch type with obesity, overweight and obesity and central obesity.

## Data Availability

Data are available at https://knhanes.cdc.go.kr/.

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
