# Peer review of "Associations between School Lunch and Obesity in Korean Children and Adolescents Based on the Korea National Health and Nutrition Examination Survey 2017–2019 Data: A Cross-Sectional Study"

_nutrients, 2023, doi:10.3390/nu15030698_

Round 1

Reviewer 1 Report

·         Title: Please change to “Associations between School Lunch and Obesity in Korean Children and Adolescents based on the National Health and Nutrition Examination Survey 2017-19 data. A cross-sectional study”.

·         The abstract is very weak. This statement is vague and should be revised in Line 13-15. Define the type of study design in Line 15-18 (cross-sectional/longitudinal). The selected groups should be clearly defined in Line 18-21. Please provide details about type of analysis used and significant p-value. Please add a conclusion that proposes a clear direction for future studies.

·         Please add “Korea” to the keywords list.

·         The novelty of this study is very weak. What is new? How this study extends reader understanding of the topic? The authors provided few references to support the aim, with only 11 studies were used. Were they cross-sectional or longitudinal? Were they conducted specifically with younger or older age groups? What are the prevalence of obesity and factors associated with in Korean school children? In general, I would suggest refining and rephrasing the whole introduction using more studies.

·         Line 56-74: Please provide more details about the recruitment of children. How they were selected? How many schools? Were schools in big cities, small town or remote villages? Were they public or private? Were they classified as low, middle or high SES? What were the exclusion criteria of the study? A diagram that shows the final selection of children is needed.

·   Line 76-82: How these variables were assessed? The use of a questionnaire/survey should be explained in greater depth, as well as justifying their use. Was it validated?

·         Line 99-100: Data collection procedure should be described in much more details.

·         How many 24-h dietary recalls were used? Was it a true 24 hour recall or 3 pre-coded questions on the overall dietary intake in the previous 24 hours? Note that one 24 hour recall may not be representative of an individual's intake and should thus not be used in the present analysis.

·      Line 107,110: Please define these variables. Did the authors check for normality distribution?

·         Line 112-113: This is not clear to me. Did you mean linear/multivariate or logistic regression analysis? I haven’t observed any results that represent multi-linear regression analysis.

·         Line 116-119: Clear rational why all these models were used?

·         It is unclear from all tables the type of analyses used?

·         Line 168-175: Please do not repeat results here.

·       Discussion: Please refer to my comment in introduction. Please define all previous studies used. Were they cross-sectional or longitudinal? Also, more recent studies to discuss the results are needed.   

·         Limitations of the study should be described in much more details.

·         How can childhood obesity be prevented from a policy level? What is the role of government? Please provide some reflection on whole of government and/or intersectoral recommendations to reduce obesity in the discussion. It would be useful to have drawn reference to childhood obesity policies from other countries (Br J Nutr. 2011 Aug;106(4):472-4; Children (Basel). 2018 Jan 29;5(2):18; Int J Med Inform. 2017 Jul;103:83-88; J Sch Health. 2014 Jun;84(6):355-62; Am J Prev Med. 2019 Jan; 56(1): e1–e11). 

·         Line 262: This is a very weak conclusion. Please provide information about the present analyses and the implications of this study and future research directions. 

·         Ref# 10,19,24,37: Very old-please delete/update.

·         Ref # 28: No need to referring to animal studies-please delete.

·    Ref # 22,29,30,33: Make sure all references are relevant (e.g., obese men/women, elderly)-please delete.

·         The manuscript requires significant editing by a native English speaker.

Reviewer 2 Report

This paper reports the results of an analysis of the Korean NHANES national survey examining the association between school lunch consumption and children’s daily dietary intake and weight status.  Strengths of the study involve the large, nationally representative sample; the use of multiple measures of weight status; the assessment of physical activity; and appropriate statistics.   Limitations involve the reliance on a single day to assess dietary intake and the cross-sectional design.  The authors found numerous differences between children who had eaten a school lunch the day before data collection and those who didn’t in the quality of their dietary intake; the only differences in weight status were between children who had skipped lunch versus others.

The procedures are well-described and the statistics appropriate.  Several issues, if addressed, would make for a much stronger paper:

First, it is difficult to understand how this paper contributes to the current literature in this area due to numerous inconsistencies in descriptions of previous research.  On lines 45 – 49 in the introduction, it is stated that research on the association between school lunch and childhood obesity/dietary intake in the U.S. is inconsistent (citing reference numbers 6 – 11). However, on lines 180 – 183 in the discussion it is stated that “However, to date, most studies have shown that SL increases the risk of obesity even if there is lack of the evidence [6,15]. SL increased the risk of obesity because the SL group was more likely to take excess fat and saturated fat, and other nutrients [16,17].”  I am not that familiar with the literature in this area, but a quick literature search on my part revealed several studies not cited here, including the following:

Gunderson et al. (2012).  The Impact of the National School Lunch Program on Child Health:  A Nonparametric Bounds Analysis.  Journal of Econometrics, 166, 79-91.

 I would encourage the authors to go back to the literature and make sure that they have identified the relevant studies, as well as present a more clear and consistent presentation of this literature.  Since at least one of the studies they cite is from Japan [reference number 11], so they should make clear that they acknowledge where these studies were done since (as they state) the school lunch programs vary considerably across different countries.

 Second, the authors conclude the article with the following statement:  “Therefore, having the school lunch may help prevent obesity among children and adolescents in Korea.”  However, one really can’t draw this conclusion from these findings since the obesity and overweight rates did not differ for the school and school lunch groups; they only differed for the lunch skipping versus the other groups.  There may be many other unmeasured lifestyle differences between children who do and don’t skip lunch that might account for these differences in obesity rates.  Therefore, it seems to me that conclusion of this paper should read that children who consume lunches have healthier diets than those who don’t, but that childhood obesity is only related to lunch skipping.  In terms of childhood obesity, there is little evidence that school lunch consumption in this sample was associated with a less healthy weight status and that future research should identify some of the possible factors that might contribute to the differences in obesity levels between children who do and don’t skip lunch.   

 Finally, I found one minor error in the discussion—on line 171 it is reported that children in the SL group had higher intakes of retinol than children in the other groups.  In Table 3 it is reported that there were no significant group differences on this variable. 

Round 2

Reviewer 1 Report

No further comments.